# The Effects of Beta-Blockers on Leukocytes and the Leukocyte Subpopulation in Heart Failure Patients

**DOI:** 10.3390/biomedicines12122907

**Published:** 2024-12-20

**Authors:** Anca Daniela Farcaş, Mirela Anca Stoia, Diana Larisa Mocan-Hognogi, Cerasela Mihaela Goidescu, Alexandra Florina Cocoi, Florin Petru Anton

**Affiliations:** 1Internal Medicine Department, Faculty of Medicine, “Iuliu Hatieganu” University of Medicine and Pharmacy Cluj-Napoca, 400012 Cluj-Napoca, Romania; ancafarcas@yahoo.com (A.D.F.);; 21st Cardiology Department, Cluj-Napoca Emergency County Hospital, 400347 Cluj-Napoca, Romania; 3Department of Cardiology, “Constantin Papilian” Emergency Military Hospital, 400132 Cluj-Napoca, Romania; 4Monza Ares Hospital, 400347 Cluj-Napoca, Romania; alexandraflorina.popa@gmail.com

**Keywords:** beta-blockers, cardiovascular drug, heart failure, neutrophil–lymphocyte ratio, leukocyte subpopulation

## Abstract

**Background/Objectives:** Some specific types of white blood cells (WBCs) and the neutrophil/lymphocyte ratio (NLR) are independent predictors of outcome for heart failure (HF) patients. WBC redistribution is induced by catecholamines, and therefore we evaluate how different types of beta-blockers (BBs) influence it. **Methods:** The HF patients were clinically evaluated, and blood was drawn to measure N-Terminal pro–B-type natriuretic peptide (NT-proBNP), WBC-differential formula, etc. **Results:** On admission, 61.16% of patients who used a BB had no significant difference in the number of lymphocytes (Lym) and neutrophils (Neu), but NLR and NT- proBNP were significantly lower compared with those without BB. NT-proBNP correlated with BB dose on admission and was significantly lower in patients treated with Metoprolol (Met) as compared with Carvedilol (Car). The type and dose of BB used was responsible for 6.1% and 5.9% of the variability in the number of Lym and Neu, respectively. Patients treated with ≥100 mg Met/day had a higher Lym number, but not of Neu, with reduced NLR, compared with lower doses. Patients treated with ≥25 mg Car/day had a lower Lym number and a greater Neu number, compared with lower doses, with increased NLR. **Conclusions:** However, both BBs had the same rehospitalization rate during the 12 month follow-up and had an improved outcome.

## 1. Introduction

Heart failure (HF) is still associated with an increased mortality and rehospitalization rate in developed countries [1]. Neuro-hormonal disturbances in HF cause disorders of other systems through the activation of inflammation and the immune system [2]—key mechanisms for the progression of HF. Hence, enhanced catecholamine serum levels, (WBC formula)—such as lymphocytes (Lym) and granulocytes (Neu)—stimulate the immune system [3] and cause a significant redistribution of the WBC formula. Also, the levels of circulating cytokines are higher in patients with dilative cardiomyopathy and HF compared with controls [4].

The neutrophil–lymphocyte ratio (NLR) is a simple yet powerful marker of systemic inflammation, calculated by dividing the number of neutrophils by the number of lymphocytes in the blood. It has been increasingly recognized as a valuable prognostic indicator in various cardiovascular conditions, including myocardial infarction (MI) and HF [5]. In patients with HF, both preserved ejection fraction heart failure (HFpEF) and reduced ejection fraction heart failure (HFrEF), the number of lymphocytes and the NLR are predictors of intra-hospital and after-discharge mortality [5], but also of the event-free interval until rehospitalization in patients with ischemic HF [6].

In HF, particularly heart failure with reduced ejection fraction (HFrEF), chronic systemic inflammation is a key component of disease progression. An elevated NLR in HF patients is indicative of this ongoing inflammatory state. Similar to MI, a higher NLR in patients with HF is associated with poorer outcomes, including higher mortality rates, increased hospitalizations, and worse functional status. The chronic elevation in NLR reflects the persistent immune activation and inflammatory burden that characterizes HF. Studies have shown that NLR correlates with the severity of HF, with higher ratios observed in more advanced stages of the disease. This makes it a useful marker for assessing disease progression and guiding treatment decisions [7,8,9,10]. In summary, the NLR is a valuable marker in both MI and HF, providing insight into the inflammatory state of the patient and helping predict clinical outcomes. Its simplicity and prognostic power make it an important tool in the management of these cardiovascular conditions [11].

Even though the available agents have different action mechanisms, BBs’ benefit in patients with HF resides in the inhibition of excessive chronic stimulation of AR, with a positive impact on the renin–angiotensin–aldosterone system (RAAS) and possibly on the immune system [12].

In the class of beta-blockers, four have demonstrated significant mortality and morbidity benefits for certain patients with congestive heart failure. Among these, metoprolol and bisoprolol are noted for their high selectivity for beta-1 AR. Carvedilol stands out due to its unique combination of beta-1, beta-2, and alpha-1 blocking effects. A large randomized clinical trial indicated that carvedilol may provide additional prognostic advantages for congestive HF patients when compared to those receiving metoprolol [13]. BBs might reduce neutrophil activity, including their ability to produce reactive oxygen species (ROS). Recent research, using various methods, has shown that beta-blockers can reduce oxidative stress and the formation of reactive oxygen species (ROS) in neutrophils, primarily through two mechanisms: inhibiting catecholamine-induced oxidative stress and exerting direct antioxidant effects. Some beta-blockers, such as carvedilol and nebivolol, have additional direct antioxidant effects. Carvedilol, for instance, has been shown to scavenge free radicals and inhibit lipid peroxidation, thus reducing ROS levels and protecting cells from oxidative damage. Nebivolol enhances nitric oxide (NO) availability, which also helps to counteract oxidative stress by balancing ROS levels. This direct action on ROS modulation distinguishes certain beta-blockers from others, as not all have these antioxidant capabilities. Furthermore, studies indicate that beta-blockers like metoprolol and carvedilol can reduce the activity of myeloperoxidase (MPO), an enzyme in neutrophils that contributes to ROS production during inflammatory responses. By inhibiting MPO, these beta-blockers help to modulate oxidative stress directly within neutrophils, reducing inflammation and preventing excessive ROS formation. This research represents a solid background and is concordant with the hypothesis that the beneficial effects of betablockers in heart failure and myocardial infarction patients is partly due to immune modulation and the reduction in oxidative stress, both in the myocardium and the blood [14,15,16,17].

This could potentially suppress certain immune responses, including cytokine production [18]. BBs may alter the balance of lymphocyte subsets. For example, they might reduce the proportion of T-helper cells (CD4+) relative to other types of lymphocytes. In patients with conditions like HF or MI, BBs may reduce inflammation and oxidative stress, which can affect leukocyte counts and activity. However, this effect might vary depending on the specific BB and the patient’s condition [18].

The β1-adrenergic receptor antagonist metoprolol (Met) reduces infarct size in the acute phase of myocardial infarction patients by acting not only on cardiomyocytes but also on the hematopoietic system. Met inhibits neutrophil migration through a β1-adrenergic receptor-dependent mechanism. It acts during the early stages of neutrophil recruitment by disrupting the structural and functional changes necessary for effective interaction with circulating platelets, leading to irregular intravascular behavior and reduced inflammation [19]. But in a post hoc analysis of the OBTAIN study, there was no difference in survival rate after myocardial infarction between Met- and Car-treated patients [20]. Also, it was shown that β1-selective blockade with Met (but not esmolol) disrupts inflammatory responses induced by TNF-α and the subsequent cytokine production [21]; additionally, Met limits neutrophil activity during heightened inflammation following myocardial infarction, providing protection against reperfusion injury [22].

Nebivolol also exhibits strong beta-1 adrenoreceptor selectivity. In addition, nebivolol is known to promote endothelial nitric-oxide-mediated vasodilation. Experimental studies have suggested that nebivolol may possess anti-inflammatory properties, as demonstrated in vitro, where nebivolol reduced the expression of pro-inflammatory genes in endothelial and vascular smooth muscle cells [23].

In summary, while beta-blockers primarily affect the cardiovascular system, they can have secondary effects on the immune system, particularly in terms of modulating inflammation. They can suppress overactive immune responses, reducing the migration and activation of these cells in inflamed tissues, and have a beneficial effect on the clinical development of some cardiovascular diseases.

The aim of our study was to investigate the effect of BB selectivity on the sympathetic stimulation of WBC subpopulations and the NLR in patients with HF.

## 2. Materials and Methods

We conducted a prospective study that enrolled 330 consecutively hospitalized HF patients. The patients were admitted to the cardiology ward of a County Emergency Hospital in Romania during the period of 2014–2019. The diagnosis of heart failure was made after clinical evaluation, NT-proBNP determination and echocardiographic examination. We excluded patients with conditions or treatments that influence the WBC number and differential formula—acute or chronic inflammatory states or infections, malignancies, acute coronary syndrome, peripheral and carotid ischemia, connective tissue diseases, vasculitides, etc. All patients were informed about study procedures and signed the informed consent. The study was approved by the Ethics Committee of the Iuliu Hațieganu University of Medicine and Pharmacy, Cluj-Napoca (No. 377/03NOV2014). After signing their informed consent, all patients were clinically evaluated and blood was drawn to measure NT-proBNP, total WBC, WBC-differential formula and NLR using a Beckman Coulter AU 680 device (Beckman Coulter, Inc., Brea, CA, USA). NLR was computed as the ratio of Neu to Lym. Clinical and demographic data were collected from the patients’ charts of the ongoing hospital admission, after trial enrollment. Patients were followed for one year for cardiovascular events and rehospitalizations. The primary endpoint for our study was defined as readmission of the patients for heart failure or myocardial ischemia. Patients were divided into two groups—with and without BBs on admission. Patients on BBs were further analyzed based on the type of BB therapy (Figure 1).

Statistical data processing was performed with SPSS Statistics V26 program (SPSS Inc.: An IBM Company, Chicago, IL, USA). Group comparison was performed using the chi-square test for categorical variables, Student’s test for continuous variables with normal distribution and the Mann–Whitney U test for continuous variables with abnormal distribution and ordinal variables. A value of *p* < 0.05 was deemed significant; confidence intervals were calculated for *p* = 0.05 as the threshold.

The study was conducted according to the World Medical Association Declaration of Helsinki and was approved by the Institutional Ethics Committee.

## 3. Results and Discussions

The patients admitted in our study were all diagnosed with HFrEF, in NYHA class III-IV (64.5 ± 14.5 years), and 52.12% were male. At the moment of admission, all patients were treated with metoprolol succinat (Met) (73.34%), associated with loop diuretics (100%), angiotensin-converting enzyme inhibitors (ACEI) (92.7%), mineralocorticoid receptor antagonist (MRA) (92.1%), digoxin (19.69%), statin (40%) and antiplatelets (23.6%).

On admission, 73.34% of all patients had BB treatment: 41.82% with Met, 18.48% with Carvedilol (Car), 5.46% with Nebivolol (Neb) and 7.57% with Bisoprolol (Bis). Some of the patients were intolerant to betablockers due to low blood pressure values or bradycardia.

Table 1 shows that patients with BBs on admission had significantly lower values of NT-proBNP and NLR, respectively, and higher levels of lymphocytes and basophils when compared with patients without BB treatment. There were no statistically significant differences between the number and type of the other leukocyte subpopulations.

Patients receiving beta-blockers at admission were more often women and had a higher prevalence of ischemic or non-ischemic dilated heart disease, whereas hypertensive heart disease was more common among patients not on beta-blockers. Additionally, ischemic heart disease patients were more frequently treated with Met, while Car was more commonly prescribed for those with non-ischemic dilated heart disease (Table 2).

Furthermore, patients treated with Metoprolol had lower LDL cholesterol (low density lipoprotein cholesterol) levels. All ischemic patients received statin treatment. In the non-ischemic group, treatment was tailored according to cardiovascular risk and associated vascular conditions. An important observation is that the LDL levels were not significantly different between the two patient groups, highlighting the relative uncommonality.

The NLR was increased above 4.75 more frequently in those with dilated cardiomyopathy (54/136), old MI (49/120) and diabetics and hypertension (46/116).

In our study, Met was used more frequently for patients with ischemic heart disease, while Car was more prevalent for hypertensive heart disease and dilated cardiomyopathy. Although not statistically significant, the NLR was higher in patients on Car compared to those on Met (Table 2).

Although research on a definitive cutoff value for NLR is still inconclusive, values above 4 in oncological diseases and above 5.95 in SARS-CoV-2 infections have been cited as having some predictive value for mortality. In our patients, a value more the 4.75 was found more frequently in patients with dilative heart disease (54 out of 136 patients) and old myocardial infarction (49 out of 120 patients) [24,25].

In our study, the type and dose of BBs used was responsible for 6.1% and 5.9% of the variability in the Lym and Neu number, respectively.

All patients treated with β1-selective BBs (Met, Bis and Neb) showed increased Neu and decreased NT-proBNP levels compared with those without BB treatment. The effect on the leukocyte number and formula is variable—patients on Met had lower values of NLR and higher levels of monocytes, basophils and eosinophils, while those on Bis and Neb showed only an increased Lym number.

Patients treated with Car had significantly decreased levels of NT-proBNP and eosinophils and increased NLR, as compared with those without BBs. Instead, when compared with those treated with Met, patients on Car had increased Lym and NT-proBNP levels, but lower values of Neu, with a greater NLR (Table 1).

Moreover, patients treated with at least 25 mg Car had a smaller amount of Lym and a greater amount of Neu compared with those treated with lower doses, with a significant rise in the NLR value. In contrast, patients who received treatment with at least 100 mg Met/day displayed a significant rise in the Lym number, with a significant change in the amount of Neu compared with those treated with lower doses, with a significantly lower NLR (Table 3).

When comparing ischemic and non-ischemic heart failure etiologies, the ischemic group showed a significantly higher value of NLR (Table 4), along with elevated C-reactive protein levels, total leukocyte and neutrophil count.

There were no significant changes in the WBC formula and NLR in patients treated with Neb and Bis, regardless of the dose used. Serum NT-proBNP levels correlated negatively with the BB dose and were significantly lower in patients receiving Met compared with those on Car (Figure 2 and Figure 3). Regardless of the type of BBs used, the rehospitalization rates during the 12 months of follow-up were similar.

Previous studies have shown that HF patients show a decrease in the amount of Lym [13], associated with a significantly modified distribution of WBC differential formula, especially Neu and Lym subsets [3].

The changes in the WBC differential formula (reduction in the Lym number and increased Neu number) induced by the chronic sympathetic stimulation are more pronounced in patients without BBs, whereas BBs seem able to reverse the changes in the leukocyte distribution observed in these patients [3].

In our study, patients with BBs had significant changes only in Lym but not Neu. A possible explanation could be the AR-selectivity of the type of BB used—Met (41.82% in our study) vs. Car (59%—in the study of Haehling et al.) [26].

Furthermore, the β-1 selectivity of Met decreases as the dose increases, and thus Met becomes non-selective in higher doses, while Car is non-selective regardless of the dose [27]. The differences in the response of Leu subpopulations to adrenergic stimulation are caused by the significant differences in the number, type and affinity of AR [28]—Lym have a higher density of β-2-AR, while the increase in Neu numbers is caused by α-AR stimulation [9]. Regardless of the effects of BB selectivity and dosage on Leu numbers and distribution, there were no significant differences in outcome and rehospitalizations between patients, a result that might be explained by another BB-specific effects (e.g., NT-proBNP).

Despite our efforts to investigate the specific effects of BBs on the hematological profile, our patients were receiving optimal medical therapy for HF, which included multiple concomitant medications. RAAS inhibition is known to exert anti-inflammatory effects, potentially reducing neutrophil recruitment and activation. Additionally, by modulating the RAAS, ACEI can influence the immune response and alter the expression of adhesion molecules on neutrophils, affecting their function and lifespan [29]. Therefore, the observed changes in the hematological profile may result from complex interactions within the neurohormonal treatment regimen, and also from the etiology of heart failure.

Numerous studies have investigated the NLR in relation to ischemic heart disease (IHD), focusing on its value as a prognostic marker in both acute coronary syndrome (ACS) and chronic coronary artery disease (CAD). These studies consistently found that NLR is an independent predictor of mortality in patients with non-ST elevation myocardial infarction (NSTEMI), correlating with the severity of the inflammatory response to ischemia, thus establishing its utility as a prognostic marker in IHD. The significance of NLR as an indicator of systemic inflammation affecting the prognosis of IHD, especially in non-ST elevation events, was underscored [30,31].

Prior research indicates that the function, structure and quantity of adrenergic receptors can be affected by various physical conditions, including differences between healthy and pathological states, as well as the acute and chronic nature of a disease and biological age, as shown by Vida et al. in Figure 4 and by Kyriazis in Figure 5. Consequently, the extent to which adrenergic stimulation influences immune responses, inflammation and oxidative stress—and by extension how betablockers impact the NLR—is intricate and multi-faceted [32].

Additionally, NLR was identified as an independent predictor of major adverse cardiovascular events (MACE) and mortality. It was validated as a prognostic tool across a diverse population of acute coronary syndrome patients [35].

In our study, the changes in hematologic patterns aligned with previously published findings, with Met-treated patients showing a lower NLR compared to those treated with other BBs, suggesting that metoprolol may have a unique effect on neutrophils rather than a class-wide effect.

The primary limitations of our study lie in the small sample size and the lack of representation for all beta-blockers used in heart failure treatment. Additionally, the patient groups treated with different beta-blockers were uneven in size, with the carvedilol group being less represented than the metoprolol group. All patients were on optimized heart failure treatment, including ACEI and MRA, so we cannot attribute changes in the NLR solely to the effects of beta-blockers. Another limitation of our study is the inability to assess oxidative stress levels in our patients. According to published research, the interplay between HNE, a marker of oxidative stress, and neutrophils is intricate and context-dependent, contributing to both immune defense mechanisms and the development of inflammatory and oxidative-stress-related diseases.

## 4. Conclusions

We report that, although the type and dosage of the BBs could have different effects on the Leu number, WBC differential formula, NLR and NT-proBNP, there were no significant differences in the outcomes of HF patients. However, we believe that further trials with a higher number of patients treated with a more balanced distribution of different types of BBs could provide additional information on this subject.

## Figures and Tables

**Figure 1 biomedicines-12-02907-f001:**
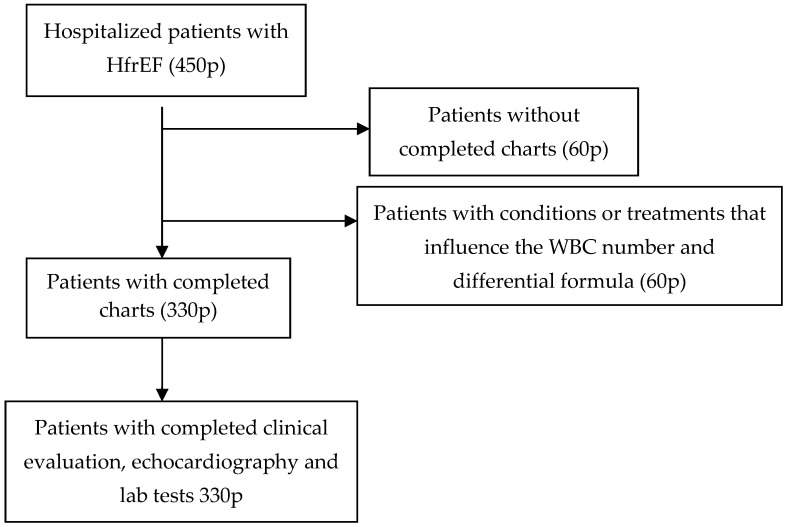
Design of the study.

**Figure 2 biomedicines-12-02907-f002:**
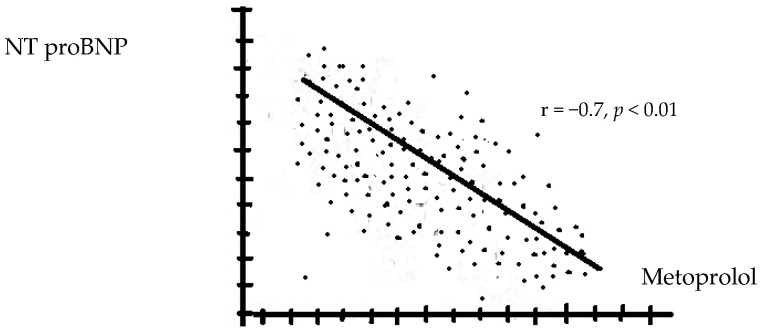
The correlations between the levels of NT pro-BNP with metoprolol (higher doses reduce the level of NT proBNP more).

**Figure 3 biomedicines-12-02907-f003:**
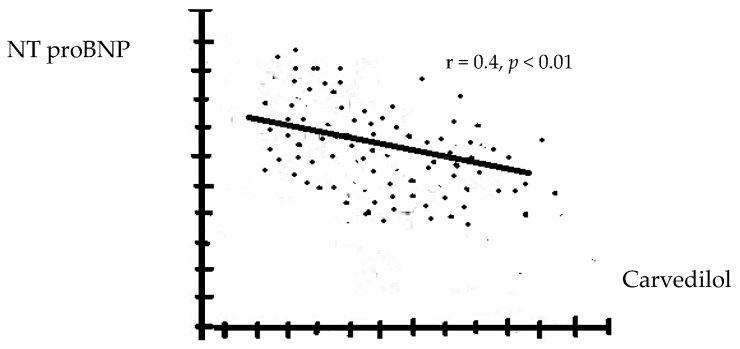
The correlations between the levels of NT pro-BNP with carvedilol (higher doses reduce the level of NT proBNP more).

**Figure 4 biomedicines-12-02907-f004:**
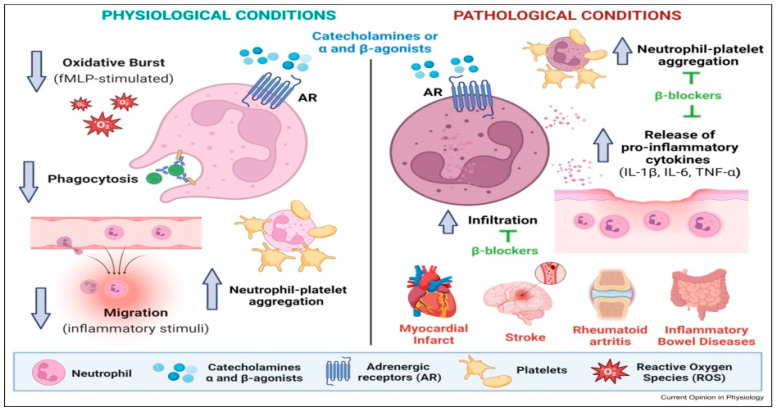
Adrenergic governance of neutrophil functions in physiology and pathology. Use of β-blockers restores neutrophil functions in diseased states [33].

**Figure 5 biomedicines-12-02907-f005:**
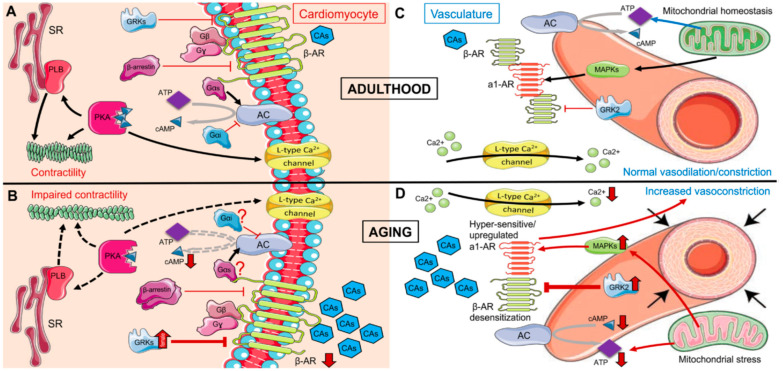
Adrenergic signaling in cardiomyocytes and vasculature during adulthood (**A**,**C**) and aging (**B**,**D**) [34].

**Table 1 biomedicines-12-02907-t001:** Baseline characteristics of patients treated with and without BBs on admission.

	Patients without BBs (88p)	Patients with BBs (242p)	*p*
Age (years)	68.41 ± 12.69	67.09 ± 15.53	0.669
Sex (men)	64.8%	35.2%	0.004
Length of stay (days)	7.45 ± 3.54	7.29 ± 3.43	0.458
Leukocytes (10^9^/L)	7.19 ± 2.20	7.84 ± 3.26	0.159
Lymphocytes (%)	14.18 ± 9.44	18.32 ± 10.59	0.019
Lymphocytes (number)	1.06 ± 0.83	1.09 ± 0.82	0.818
Neutrophils (%)	54.77 ± 29.88	53.66 ± 31.01	0.493
Neutrophils (number)	4.25 ± 3.18	4.20 ± 2.93	0.907
Monocyte (%)	3.14 ± 4.52	4.41 ± 4.18	0.763
Eosinophils (%)	1.18 ± 2.11	1.17 ± 1.55	0.963
Basophils (%)	0.06 ± 0.11	0.14 ± 0.35	0.042
NLR (neutrophil–lymphocyte ratio)	5.15 ± 8.68	3.64 ± 3.87	0.039
Ejection fraction (%)	31.49 ± 19.22	36.7 ± 18.55	0.045
Ischemic heart disease	30.6%	69.4%	0.005
Hypertensive heart disease	54.6%	45.4%	0.037
Idiopathic dilated cardiomyopathy	42.6%	58.4%	0.034
NT-proBNP pg/mL	5879.03 ± 2804.58	3283.39 ± 1079.66	0.010

**Table 2 biomedicines-12-02907-t002:** Demographic, clinical and biological characteristics of patients treated with Met versus Car.

Characteristics (Mean ± SD)	Patients with Metoprolol (138p)	Patients with Carvedilol (60p)	*p*
Age (years)	66.81 ± 15.34	64.82 ± 16.04	0.576
Sex (men)	48.4%	51.6%	0.025
Length of stay (days)	7.25 ± 4.21	8.04 ± 3.47	0.356
Leukocytes (10^9^/L)	7.64 ± 2.36	7.84 ± 2.30	0.052
Lymphocytes (%)	12.75 ± 13.09	15.35 ± 10.41	0.015
Lymphocytes (10^9^/L)	1.28 ± 0.93	0.86 ± 0.70	0.039
Neutrophils (%)	53.92 ± 28.92	37.56 ± 35.63	0.034
Neutrophils (10^9^/L)	4.55 ± 3.41	3.14 ± 3.03	0.015
Monocyte (%)	3.53 ± 4.16	2.51 ± 3.67	0.038
Eosinophils (%)	1.00 ± 1.75	1.21 ± 1.95	0.672
Basophils (%)	0.07 ± 0.10	0.06 ± 0.12	0.877
History of myocardial ischemia	76.4%	23.6%	0.004
Hypertensive heart disease	34.6%	65.4%	0.003
Non-ischemic dilated cardiomyopathy	45.2%	54.8%	0.035
NLR (neutrophil/lymphocyte ratio)	4.26 ± 3.56	4.58 ± 4.17	0.043
NT-proBNP pg/mL	1079.14 ± 283.48	4439.29 ± 2328.56	0.002
LDL-cholesterol (mg/dL)	91.8 ± 42.6	102.8 ± 53.10	0.046

**Table 3 biomedicines-12-02907-t003:** Baseline characteristics of patients treated with Met and Car on admission.

Characteristics (Mean ± SD)	Metoprolol <100 mg/Day (75p)	Metoprolol ≥100 mg/Day (63p)	*p*	Carvedilol <25 mg/Day (32p)	Carvedilol ≥25 mg/Day (28p)	*p*
Age (years)	64.83 ± 13.24	65.21 ± 14.34	0.245	62.85 ± 15.14	65.23 ± 16.27	0.576
Length of stay (days)	7.31 ± 3.82	7.55 ± 4.31	0.467	8.04 ± 3.47	7.85 ± 3.39	0.356
Leukocytes (10^9^/L)	6.50 ± 2.10	7.40 ± 2.62	0.655	7.98 ± 1.83	7.52 ± 0.30	0.470
Lymphocytes (%)	14.65 ± 3.33	16.23 ± 5.15	0.027	13.24 ± 10.35	12.52 ± 14.52	0.043
Lymphocytes (10^9^/L)	1.25 ± 0.51	1.63 ± 1.07	0.043	0.92 ± 0.76	0.73 ± 0.59	0.115
Neutrophils (%)	68.26 ± 12.96	24.66 ± 12.72	0.004	36.33 ± 35.81	40.19 ± 37.93	0.807
Neutrophils (10^9^/L)	3.99 ± 2.68	4.67 ± 2.23	0.023	2.29 ± 2.10	4.60 ± 3.94	0.005
Monocyte (%)	3.53 ± 3.16	3.79 ± 2.73	0.052	1.48 ± 0.81	1.72 ± 0.51	0.051
Eosinophils (%)	0.87 ± 0.67	0.98 ± 0.82	0.372	0.96 ± 0.82	1.75 ± 0.85	0.294
Basophils (%)	0.6 ± 0.10	0.7 ± 0.15	0.056	0.16 ± 0.11	0.8 ± 0.69	0.725
NLR (neutrophil/lymphocyte ratio)	4.85 ± 2.67	3.18 ± 2.53	0.029	2.46 ± 1.09	7.10 ± 4.41	0.014
NT-proBNP (pg/mL)	1249.41 ± 320.17	674.88 ± 220.15	0.414	5094.81 ± 2786.43	2351.71 ± 1282.00	0.046

**Table 4 biomedicines-12-02907-t004:** Baseline characteristics of patients with non-ischemic heart failure (NIHF) versus ischemic heart failure (IHF).

	Patients with NIHF	Patients with IHF	*p*
Age	62.08 ± 12.58	63.02 ± 14.28	NS
Women	64.6%	63.9%	NS
Men	36.4%	37.1%	NS
Hemoglobin (g/dL)	12.10 ± 3.47	13.25 ± 3.56	NS
Hematocrit (%)	34.28 ± 13.41	32.12 ± 13.58	NS
Leukocytes (10^9^/L)	6.23 ± 1.98 × 10^9^	7.36 ± 3.26 × 10^9^	<0.001
Neutrophils (10^9^/L)	51.5 ± 32.6	56.6 ± 24.8	<0.005
Lymphocytes (10^9^/L)	14.2 ± 11%	16.2 ± 9.2%	<0.05
NLR (neutrophil–lymphocyte ratio)	0.89 (0.28–1.05)	1.17 (1.12–1.43)	<0.02
CRP (mg/dL) (CRP: C reactive protein)	0.14 ± 0.34	0.28 ± 0.58	<0.001

## Data Availability

The original contributions presented in the study are included in the article, further inquiries can be directed to the corresponding author.

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
