# Peer review of "The Effects of Beta-Blockers on Leukocytes and the Leukocyte Subpopulation in Heart Failure Patients"

_biomedicines, 2024, doi:10.3390/biomedicines12122907_

Round 1

Reviewer 1 Report

Comments and Suggestions for Authors

The results paragraph has to be completed: patient population is described insufficiently - what kind of HF (HFrEF?), other medications beyond BB?, metoprolol succinate?

The term idiopatic DCM I would change to simple DCM or non-ischemic DCM, as etiological correlations were not studied.

Unfortunately there is a heterogenity between the two BB populations (e.g. severity, etiology) which could account at least partly for the results. Please comment on this aspect. 

I would like not used the term "BB used have different effects", because it is hard to demonstrate, that only the BB treatment is responsible for the results.

Tabel 4 has errors, the values of NLR are not realistic, limphocites, etc.

Comments on the Quality of English Language

ok

Author Response

Comment 1 The results paragraph has to be completed: patient population is described insufficiently - what kind of HF (HFrEF?), other medications beyond BB?, metoprolol succinate?

Answer1 . All patients had a left ventricular ejection fraction below 40%, classifying the group as having heart failure with reduced ejection fraction (HFrEF). All patients were treated with Metoprolol succinat (73,34%), associated with loop diuretics (100%), ACEI (92,7%), MRA (92,1%), Digoxin (19,69%), statin (40%), and antiplatelets (23,6%).

Comment 2 .The term idiopatic DCM I would change to simple DCM or non-ischemic DCM, as etiological correlations were not studied.

Answer 2 We have, accordingly, changed the terms used in text.

Comment 3 .  Unfortunately there is a heterogenity between the two BB populations (e.g. severity, etiology) which could account at least partly for the results. Please comment on this aspect. 

Answer 3. Our patient population includes individuals with HFrEF from various causes, all of whom were admitted for heart failure classified as NYHA III-IV. We aimed to analyze a non-invasive, easily reproducible, and readily available laboratory test and its predictive value for patient outcomes.

Comment 4.  I would like not used the term "BB used have different effects", because it is hard to demonstrate, that only the BB treatment is responsible for the results.

Answer 4.  We agree with this comment and the modification was made.

Comment 5. Tabel 4 has errors, the values of NLR are not realistic, limphocites, etc.

Answer 5. The typing errors were corrected. The values in the table are correct, calculated from the patients’ data.

Reviewer 2 Report

Comments and Suggestions for Authors

This manuscript investigated the effect of beta-blockers on leukocytes and the leukocytes  sub-population in heart failure patients. The results may be interesting to the clinical doctors. I would like to have some suggestions:

1. The numbers of patients involved in each group should be listed in the tables. Although the author indicated 300 patients were involved in the overall study, the detail information should be included in the table.

2. It is better to prepare a figure to illustrate the procedure or design of the study to help the reader to understand.

3.   The authors indicated that NT-proBNP correlates with BB dose. Could the authors used a figure to show the correlations? Only data in Table were not sufficient.

Author Response

Comment 1. The numbers of patients involved in each group should be listed in the tables. Although the author indicated 300 patients were involved in the overall study, the detail information should be included in the table.

Response 1. The tables were amended.

Comment 2. It is better to prepare a figure to illustrate the procedure or design of the study to help the reader to understand.

Response 2. Agree. We realise a figure to illustrate the procedure or design of the study ..

Comment 3. The authors indicated that NT-proBNP correlates with BB dose. Could the authors used a figure to show the correlations? Only data in Table were not sufficient.

Response 3. We realize graphs to highlight the results better.

Reviewer 3 Report

Comments and Suggestions for Authors

Clinical assessment of patients with heart failure is very relevant and interesting but I have some important remarks.

1. The title and aim of the manuscript indicate heart failure as an area of ​​interest, but the introduction and discussion contain a lot of information about myocardial infarction, so the title does not fully match the content. 2. The Materials and Methods section contains information about end points and statistical methods that are not further described in the text. 3. The study is described as prospective, however “Clinical and demographic data were collected from the patients’ admission charts” 4. The difference between the groups according to LDL is indicated, but no information is provided on the use of lipid-lowering drugs (Line 153). 5. The text data contradicts the tables (lines 181-186 and Table 3). 6. There is no reference to Table 4 in the text. 7. It is stated that the treatment of CH was optimal, however, more than 25% of patients did not take BB without explanation. 8. The country of manufacture of the biochemical analyzer and the country of origin of the statistics are not indicated. 9. Sources: 13 (out of 30) are older than 10 years, 2 self-citations.

10. Please check for typos and use of abbreviations.

Author Response

Comments 1.  The title and aim of the manuscript indicate heart failure as an area of ​​interest, but the introduction and discussion contain a lot of information about myocardial infarction, so the title does not fully match the content.

Response 1. The introduction of our article aims to connect the intricate relationship between acute and chronic ischemia and the progression of heart failure, as well as the alterations in leukocyte activity and patterns. Given that our patient population includes both ischemic and non-ischemic heart failure, we deemed the relationship between these conditions significant for our research.

Comment 2. The Materials and Methods section contains information about end points and statistical methods that are not further described in the text.

Response 2. The primary enpoint was the number of readmissions. We mentioned in the Results that “Regardless of the type of BB used, the rehospitalization rates during the 12 months of follow-up were similar.”

Comments 3. The study is described as prospective, however “Clinical and demographic data were collected from the patients’ admission charts” 

Response 3. The study was prospective: during the initial patient evaluation and trial enrollment, data from the ongoing hospitalization records were collected and documented. The patients were then followed for a year to monitor events.

Comment 4. The difference between the groups according to LDL is indicated, but no information is provided on the use of lipid-lowering drugs (Line 153).

Response 4. All ischemic patients received statin treatment. In the non-ischemic group, treatment was tailored according to cardiovascular risk and associated vascular conditions. An important observation is that the LDL levels were not significantly different between the two patient groups, highlighting the relatively uncommon.

Comment 5. The text data contradicts the tables (lines 181-186 and Table 3).

Response 5. We reviewed the data and found no discrepancies.

Comment 6. There is no reference to Table 4 in the text.

Response 6. The text was amended.

Comment 7. It is stated that the treatment of CH was optimal, however, more than 25% of patients did not take BB without explanation.

Response 7. Some of the patients were intolerant to betablockers due to the low blood pressure values or bradycardia.

Comment 8. The country of manufacture of the biochemical analyzer and the country of origin of the statistics are not indicated.

Response 8. The blood analyses were done on a Beckman-Coulter AU 680 device, property of Emergency County Hospital of Cluj-Napoca. The statistical analyses were made using SPSS 16 program ….

Comment 9. Sources: 13 (out of 30) are older than 10 years, 2 self-citations.

Response 9. All sources were carefully chosen. Although they are over 10 years old, we believe they are still relevant to our research background.

Reviewer 4 Report

Comments and Suggestions for Authors

The reviewer thanks the authors for their interesting submission. A few preliminary queries:

1. Could the authors provide reference(s) to support the claim in line 77: "BBs might reduce neutrophil activity, including their ability to produce reactive 77 oxygen species (ROS)." Given this seems to be the central/main concept of the manuscript.

2. The authors should consider creating some illustrations to complement their article, and could convert some of the tables into scatter with mean+SEM plots for improved communication of their findings.

Comments on the Quality of English Language

Misspelled words are scattered throughout the article, likely secondary to a subset of the English dialect.

Author Response

Comment 1.  Could the authors provide reference(s) to support the claim in line 77: "BBs might reduce neutrophil activity, including their ability to produce reactive 77 oxygen species (ROS)." Given this seems to be the central/main concept of the manuscript.

Response 1. Nakamura K, Murakami M, Miura D, Yunoki K, Enko K, Tanaka M, Saito Y, Nishii N, Miyoshi T, Yoshida M, Oe H, Toh N, Nagase S, Kohno K, Morita H, Matsubara H, Kusano KF, Ohe T, Ito H. Beta-Blockers and Oxidative Stress in Patients with Heart Failure. Pharmaceuticals (Basel). 2011 Aug 5;4(8):1088-100. doi: 10.3390/ph4081088. PMID: 26791643; PMCID: PMC4058661.

Comment 2.The authors should consider creating some illustrations to complement their article, and could convert some of the tables into scatter with mean+SEM plots for improved communication of their findings.

Response 2.  We agree that making some graphs could help to understand the results. Please help us by specifying which of the parameters would be more important and should be included in the graphs.

Round 2

Reviewer 1 Report

Comments and Suggestions for Authors

All the issues raised in the review report were properly treated by the authors.

Comments on the Quality of English Language

none

Author Response

Thank you. Best regards, 

Anca Farcas

Reviewer 3 Report

Comments and Suggestions for Authors

Comments 1. The title and aim of the manuscript indicate heart failure as an area of ​​interest, but the introduction and discussion contain a lot of information about myocardial infarction, so the title does not fully match the content.

Response 1. The introduction of our article aims to connect the intricate relationship between acute and chronic ischemia and the progression of heart failure, as well as the alterations in leukocyte activity and patterns. Given that our patient population includes both ischemic and non-ischemic heart failure, we deemed the relationship between these conditions significant for our research.

Comments 1 (29Oct2024). The pathogenesis of acute myocardial infarction and heart failure is different; therefore, it is incorrect to compare the dynamics of  leukocytes levels in these diseases. Please delete information about AMI, especially since the introduction takes 2 pages (that’s a lot).

Comment 2. The Materials and Methods section contains information about end points and statistical methods that are not further described in the text.

Response 2. The primary enpoint was the number of readmissions. We mentioned in the Results that “Regardless of the type of BB used, the rehospitalization rates during the 12 months of follow-up were similar.”

Comment 3. The study is described as prospective, however “Clinical and demographic data were collected from the patients’ admission charts”

Response 3. The study was prospective: during the initial patient evaluation and trial enrollment, data from the ongoing hospitalization records were collected and documented. The patients were then followed for a year to monitor events.

Comment 4. The difference between the groups according to LDL is indicated, but no information is provided on the use of lipid-lowering drugs (Line 153).

Response 4. All ischemic patients received statin treatment. In the non-ischemic group, treatment was tailored according to cardiovascular risk and associated vascular conditions. An important observation is that the LDL levels were not significantly different between the two patient groups, highlighting the relatively uncommon.

Comments 4 (29Oct2024). Please, add this information in manuscript.

Comment 5. The text data contradicts the tables (lines 181-186 and Table 3).

Response 5. We reviewed the data and found no discrepancies.

Comment 5 (29Oct2024). “In contrast, patients who received treatment with at least 100 mg Met/day displayed a significant rise in the Lym number, without a significant change in  the number of Neu compared with those treated with lower doses” but in Table 3 p= .004 for Neu (%) and p= .023 for Neu (10^9/L)

Comment 6. There is no reference to Table 4 in the text.

Response 6. The text was amended.

Comment 7. It is stated that the treatment of HF was optimal, however, more than 25% of patients did not take BB without explanation.

Response 7. Some of the patients were intolerant to betablockers due to the low blood pressure values or bradycardia.

Comments 7 (29Oct2024). Please, add this information in manuscript.

Comment 8. The country of manufacture of the biochemical analyzer and the country of origin of the statistics are not indicated.

Response 8. The blood analyses were done on a Beckman-Coulter AU 680 device, property of Emergency County Hospital of Cluj-Napoca. The statistical analyses were made using SPSS 16 program ….

Comments 8 (29Oct2024). Generally accepted algorithm for describing devices in scientific articles is «Beckman-Coulter AU 680 (Beckman Coulter, Inc., USA, CA, Brea)» and SPSS 16 program (SPSS: An IBM Company, USA).

Comment 9. Sources: 13 (out of 30) are older than 10 years, 2 self-citations.

Response 9. All sources were carefully chosen. Although they are over 10 years old, we believe they are still relevant to our research background.

Comments 9  (29Oct2024). Please, try to find more modern sources (not 1983, 1990, 1992, 1996, 1998, 2001, 2007, 2008, 2009).

Author Response

Comments 1 (29Oct2024). The pathogenesis of acute myocardial infarction and heart failure is different; therefore, it is incorrect to compare the dynamics of  leukocytes levels in these diseases. Please delete information about AMI, especially since the introduction takes 2 pages (that’s a lot).

Response 1. We deleted the information about myocardial infarction from the introduction

Comments 4 (29Oct2024). Please, add this information ( about LDL ) in manuscript.

Response 4.1 – We added this information in manuscript. 

Comment 7. It is stated that the treatment of HF was optimal, however, more than 25% of patients did not take BB without explanation.

Response 7. Some of the patients were intolerant to betablockers due to the low blood pressure values or bradycardia.

Comments 7 (29Oct2024). Please, add this information in manuscript.

Response 7.1 We added this information in manuscript.

Comments 8 (29Oct2024). Generally accepted algorithm for describing devices in scientific articles is «Beckman-Coulter AU 680 (Beckman Coulter, Inc., USA, CA, Brea)» and SPSS 16 program (SPSS: An IBM Company, USA).

Response 8.1  We have inserted the information in the text.  

Comments 9  (29Oct2024). Please, try to find more modern sources (not 1983, 1990, 1992, 1996, 1998, 2001, 2007, 2008, 2009).

Response 9.1 I have removed older articles or replaced them with newer ones.

Reviewer 4 Report

Comments and Suggestions for Authors

Comment 1.  Could the authors provide reference(s) to support the claim in line 77: "BBs might reduce neutrophil activity, including their ability to produce reactive 77 oxygen species (ROS)." Given this seems to be the central/main concept of the manuscript.

Response 1. Nakamura K, Murakami M, Miura D, Yunoki K, Enko K, Tanaka M, Saito Y, Nishii N, Miyoshi T, Yoshida M, Oe H, Toh N, Nagase S, Kohno K, Morita H, Matsubara H, Kusano KF, Ohe T, Ito H. Beta-Blockers and Oxidative Stress in Patients with Heart Failure. Pharmaceuticals (Basel). 2011 Aug 5;4(8):1088-100. doi: 10.3390/ph4081088. PMID: 26791643; PMCID: PMC4058661.

Rebuttal 1. The reviewer appreciates the authors supplying the single reference, however no additional explanation/contextualization was provided along with the reference; no pertinent excerpt from the reference's text, nor any revisions to the submitted manuscript's text nor reference list. Furthermore, the limitations of 4-Hydroxy-2-nonenal (HNE) in assessing reactive oxygen species and subsequently translating that to neutrophil activity are not addressed. This reviewer recommends the authors improve their response.

Comment 2.The authors should consider creating some illustrations to complement their article, and could convert some of the tables into scatter with mean+SEM plots for improved communication of their findings.

Response 2.  We agree that making some graphs could help to understand the results. Please help us by specifying which of the parameters would be more important and should be included in the graphs.

Rebuttal 2. This reviewer would recommend the content of tables two and three be converted to graphical form, along with a mechanistic/stylistic illustration of the proposed therapeutic mechanism of action, and a central illustration that highlights the main finding of reduced neutrophil and/or lymphocyte count with treatment.

Comment 3. Numerous abbreviations have not be defined in both the abstract, manuscript, and tables. 

Comment 4. Please add a title and legend with all appropriate abbreviations defined, the statistical test defined, and the main finding conclusion to each table and/or figure.

Comment 5. Please add a limitations section.

Author Response

Rebuttal 1. The reviewer appreciates the authors supplying the single reference, however no additional explanation/contextualization was provided along with the reference; no pertinent excerpt from the reference's text, nor any revisions to the submitted manuscript's text nor reference list. Furthermore, the limitations of 4-Hydroxy-2-nonenal (HNE) in assessing reactive oxygen species and subsequently translating that to neutrophil activity are not addressed. This reviewer recommends the authors improve their response

The text was amended with the following paragraph and citations:

“Recent research, using various methods, showed that beta-blockers can reduce oxidative stress and the formation of reactive oxygen species (ROS) in neutrophils primarily through two mechanisms: inhibiting catecholamine-induced oxidative stress and exerting direct antioxidant effects. Some beta-blockers, such as carvedilol and nebivolol, have additional direct antioxidant effects. Carvedilol, for instance, has been shown to scavenge free radicals and inhibit lipid peroxidation, thus reducing ROS levels and protecting cells from oxidative damage. Nebivolol enhances nitric oxide (NO) availability, which also helps to counteract oxidative stress by balancing ROS levels. This direct action on ROS modulation distinguishes certain beta-blockers from others, as not all have these antioxidant capabilities. Furthermore, studies indicate that beta-blockers like metoprolol and carvedilol can reduce the activity of myeloperoxidase (MPO), an enzyme in neutrophils that contributes to ROS production during inflammatory responses. By inhibiting MPO, these beta-blockers help to modulate oxidative stress directly within neutrophils, reducing inflammation and preventing excessive ROS formation. This research represents a solid background and is concordant to the hypothesis that the beneficial effects of betablockers in the heart failure and myocardial infarction patients is partly due to the immune modulation and reduction of oxidative stress, both in myocardium and the blood.”

  • Jin, S.; Kang, P.M. A Systematic Review on Advances in Management of Oxidative Stress-Associated Cardiovascular Diseases. Antioxidants202413, 923. https://doi.org/10.3390/antiox13080923
  • Al-Kuraishy HM, Al-Gareeb AI, Mostafa-Hedeab G, Kasozi KI, Zirintunda G, Aslam A, Allahyani M, Welburn SC, Batiha GE. Effects of β-Blockers on the Sympathetic and Cytokines Storms in Covid-19. Front Immunol. 2021 Nov 11;12:749291. doi: 10.3389/fimmu.2021.749291. PMID: 34867978; PMCID: PMC8637815.
  • Chin BS, Langford NJ, Nuttall SL, Gibbs CR, Blann AD, Lip GY. Anti-oxidative properties of beta-blockers and angiotensin-converting enzyme inhibitors in congestive heart failure. Eur J Heart Fail. 2003 Mar;5(2):171-4. doi: 10.1016/s1388-9842(02)00251-9. PMID: 12644008.
  • Nakamura K, Murakami M, Miura D, Yunoki K, Enko K, Tanaka M, Saito Y, Nishii N, Miyoshi T, Yoshida M, Oe H, Toh N, Nagase S, Kohno K, Morita H, Matsubara H, Kusano KF, Ohe T, Ito H. Beta-Blockers and Oxidative Stress in Patients with Heart Failure. Pharmaceuticals (Basel). 2011; 5;4(8):1088-100. doi: 10.3390/ph4081088. PMID: 26791643; PMCID: PMC4058661.

Rebuttal 2. This reviewer would recommend the content of tables two and three be converted to graphical form, along with a mechanistic/stylistic illustration of the proposed therapeutic mechanism of action, and a central illustration that highlights the main finding of reduced neutrophil and/or lymphocyte count with treatment.

We have transformed tables 2 and 3 into graphs and they have been inserted into the text.  

We would add relevant information and two figures to exemplify the mechanisms involved and their effects.

The prior research indicates that the function, structure and quantity of adrenergic receptors can be affected by various physical conditions, including differences between healthy and pathological states, as well as the acute and chronic nature of a disease and biological age, as shown by Vida et all in figure 6 and by Kyriazis in figure 7. Consequently, the extent to which adrenergic stimulation influences immune responses, inflammation, and oxidative stress- and by extension, how betablockers impact the NLR- is intricate and multi-faceted. 

Figura 6. Adrenergic governance of neutrophil functions in physiology and pathology.                     Use of β-blockers restore neutrophil functions in diseased states [32] 

Figura 7. Adrenergic signaling in cardiomyocytes and vasculature during adulthood (a–c)                       and aging (b–d) [33].

Comment 3. Numerous abbreviations have not be defined in both the abstract, manuscript, and tables. 

We have checked and reviewed all the abbreviations, correcting and explaining all of them.

 Comment 4. Please add a title and legend with all appropriate abbreviations defined, the statistical test defined, and the main finding conclusion to each table and/or figure.

We have added a title and legend with all appropriate abbreviations defined to each table and/or figure..For example

Figure 2. The effect of metoprolol succinate dose of the leukocyte subpopulations, NLR and NT-proBNP levels. ( higher dose of metoprolol reduce de NLR)

(LOS – length of stay,Leu- leukocytes, Lym- lymphocytes, Mo- monocite, Eo - eosinophile, Ba- Basophil, NLR - neutrophil/lymphocyte  ratio, NT-proBNP- N-Terminal pro–B-type natriuretic peptide) 

 Comment 5. Please add a limitations section 

Response 5. The primary limitation of our study lies in the small sample size and the lack of representation for all beta-blockers used in heart failure treatment. Additionally, the patient groups treated with different beta-blockers were uneven in size, with the carvedilol group being less represented than the metoprolol group. All patients were on optimized heart failure treatment, including ACE inhibitors and aldosterone antagonists, so we cannot attribute changes in the neutrophil-to-lymphocyte ratio (NLR) solely to the effects of beta-blockers

Round 3

Reviewer 3 Report

Comments and Suggestions for Authors

NA

Author Response

Comments and Suggestions for Authors :NA  Thanks for the helpful review.  

Reviewer 4 Report

Comments and Suggestions for Authors

This reviewer thank the authors for their thorough revisions to the manuscript. A few minor concerns remain:

1. Though the conversion of some tabular data to bar graphs is an improvement, the current standard in our field is to include the individual data points for each metric/column in the figure, along with the mean and associated standard error of the mean. Please revise, and also include in the figure legends the description of the depicted data (individual data points along with mean with standard error of the mean).

2. Please expand the limitations section to include the previously discussed points raised by this reviewer (Furthermore, the limitations of 4-Hydroxy-2-nonenal (HNE) in assessing reactive oxygen species and subsequently translating that to neutrophil activity are not addressed.)

3. Please expand the discussion/references to include the concept of novel therapeutics currently under development that may recapitulate the the observed immunologic effect of systemic beta-blockade via local paracrine signaling:

Chinyere IR, Bradley P, Uhlorn J, Eason J, Mohran S, Repetti GG, Daugherty S, Koevary JW, Goldman S, Lancaster JJ. Epicardially Placed Bioengineered Cardiomyocyte Xenograft in Immune-Competent Rat Model of Heart Failure. Stem Cells Int. 2021 Jul 24;2021:9935679. doi: 10.1155/2021/9935679. PMID: 34341667; PMCID: PMC8325579.

Comments on the Quality of English Language

Some mentions of "lymphocyte" in the manuscript's tables have a mis-spelling, specifically "lymphocite".

Author Response

Comment 1. Though the conversion of some tabular data to bar graphs is an improvement, the current standard in our field is to include the individual data points for each metric/column in the figure, along with the mean and associated standard error of the mean. Please revise, and also include in the figure legends the description of the depicted data (individual data points along with mean with standard error of the mean).

Answer 1 . I have revised the graphs and included the mean and standard deviation.

Comment 2. Please expand the limitations section to include the previously discussed points raised by this reviewer (Furthermore, the limitations of 4-Hydroxy-2-nonenal (HNE) in assessing reactive oxygen species and subsequently translating that to neutrophil activity are not addressed.)

Answer 2. We included in the manuscript also : "Another limitation of our study is the inability to assess oxidative stress levels in our patients. According to published research, the interplay between HNE, a marker of oxidative stress, and neutrophils is intricate and context-dependent, contributing to both immune defense mechanisms and the development of inflammatory and oxidative stress-related diseases."

Comment 3. Please expand the discussion/references to include the concept of novel therapeutics currently under development that may recapitulate the the observed immunologic effect of systemic beta-blockade via local paracrine signaling:

Chinyere IR, Bradley P, Uhlorn J, Eason J, Mohran S, Repetti GG, Daugherty S, Koevary JW, Goldman S, Lancaster JJ. Epicardially Placed Bioengineered Cardiomyocyte Xenograft in Immune-Competent Rat Model of Heart Failure. Stem Cells Int. 2021 Jul 24;2021:9935679. doi: 10.1155/2021/9935679. PMID: 34341667; PMCID: PMC8325579.

The area of research suggested by the reviewer is outside our field of expertise, and we are unable to provide an informed opinion on it.

Some mentions of "lymphocyte" in the manuscript's tables have a mis-spelling, specifically "lymphocite". 

Answer : We have corrected the text according to the your recommendations.
